# Amplicon-Based Microbiome Profiling: From Second- to Third-Generation Sequencing for Higher Taxonomic Resolution

**DOI:** 10.3390/genes14081567

**Published:** 2023-07-31

**Authors:** Elisabetta Notario, Grazia Visci, Bruno Fosso, Carmela Gissi, Nina Tanaskovic, Maria Rescigno, Marinella Marzano, Graziano Pesole

**Affiliations:** 1Department of Biosciences, Biotechnology and Environment, University of Bari Aldo Moro, 70126 Bari, Italy; elisabetta.notario@uniba.it (E.N.); bruno.fosso@uniba.it (B.F.); carmela.gissi@uniba.it (C.G.); 2Institute of Biomembranes, Bioenergetics and Molecular Biotechnologies, Consiglio Nazionale delle Ricerche, 70126 Bari, Italy; g.visci@ibiom.cnr.it; 3CoNISMa, Consorzio Nazionale Interuniversitario per le Scienze del Mare, 00196 Roma, Italy; 4Postbiotica S.r.l., 20123 Milan, Italy; nina.tanaskovic@postbiotica.com; 5IRCCS Humanitas Research Hospital, 20089 Rozzano, Italy; maria.rescigno@hunimed.eu; 6Department of Biomedical Sciences, Humanitas University, 20072 Pieve Emanuele, Italy; 7Consorzio Interuniversitario Biotecnologie, 34148 Trieste, Italy

**Keywords:** metagenomics, microbiome, 16S rRNA amplicon-based sequencing, next-generation sequencing, third-generation sequencing, mock analysis

## Abstract

The 16S rRNA amplicon-based sequencing approach represents the most common and cost-effective strategy with great potential for microbiome profiling. The use of second-generation sequencing (NGS) technologies has led to protocols based on the amplification of one or a few hypervariable regions, impacting the outcome of the analysis. Nowadays, comparative studies are necessary to assess different amplicon-based approaches, including the full-locus sequencing currently feasible thanks to third-generation sequencing (TGS) technologies. This study compared three different methods to achieve the deepest microbiome taxonomic characterization: (a) the single-region approach, (b) the multiplex approach, covering several regions of the target gene/region, both based on NGS short reads, and (c) the full-length approach, which analyzes the whole length of the target gene thanks to TGS long reads. Analyses carried out on benchmark microbiome samples, with a known taxonomic composition, highlighted a different classification performance, strongly associated with the type of hypervariable regions and the coverage of the target gene. Indeed, the full-length approach showed the greatest discriminating power, up to species level, also on complex real samples. This study supports the transition from NGS to TGS for the study of the microbiome, even if experimental and bioinformatic improvements are still necessary.

## 1. Introduction

Over the years, the microbiome has become the focus of an increasing number of studies in different fields and applications, from environmental research to various interdisciplinary fields, e.g., agriculture, food science, biotechnology, bioeconomy, mathematics (informatics, statistics, modeling), plant pathology, and especially human medicine [1]. In particular, the human gut microbiome has gained attention due to its critical role in human health and disease. It functions as an additional organ in our body and has a vital role in physiology, metabolism, and immune responses, establishing a symbiotic relationship with the host [2]. Hence, the recent interest in investigating how the composition and function of the microbiome vary in response to diseases or how they may influence the onset of diseases. Moreover, this newly acquired knowledge has been used to develop new and innovative strategies for the prevention, diagnosis, and treatment of various disorders affecting human health [3]. In the past decade, metagenomics has greatly improved our understanding of the microbial communities, including prokaryotes, fungi, viruses, and protozoans, that inhabit the human body and other environments, although there is still a large fraction of uncharacterized micro-organisms that are sometimes called “microbial dark matter” [4]. According to Pérez-Cobas et al. [5], two major methods, amplicon-based and shotgun sequencing, represent valid approaches for exploring the microbiome, using high-throughput sequencing technologies [6]. These approaches are constantly evolving along with the rapid upgrade and development of new high-throughput sequencing technologies, which have progressed from second-generation sequencing or next-generation sequencing (NGS), to third-generation sequencing (TGS) technologies, shifting from short-read to long-read sequencing [7]. Considering the remarkable genome plasticity of eubacteria, where specific functions/features are associated with strain-specific genome tracts possibly originated by lateral gene transfer, the main challenge now is to achieve the most fine-grained taxonomic classification to gain a better understanding of the composition and function of the microbiome. [8,9,10,11]. To date, the amplicon-based sequencing approach (also known as DNA metabarcoding), based on the analysis of a target gene/genomic region, specific for the taxonomic domain of interest (e.g., 16S rRNA gene for prokaryotic characterization), remains the most common and cost-effective strategy with great potential for microbiome profiling. This is due to several peculiarities, such as (i) high sensitivity; (ii) less risk of host contamination related to the specificity of the target gene used; (iii) possibility of checking and reducing the presence of false positives [12,13]; (iv) availability of computational error correction tools; (v) several publicly available and user-friendly suites, like QIIME 2 [14] and Mothur [15]; and (vi) the lower cost compared to the shotgun sequencing approach. Until recently, most studies have focused on the amplification and sequencing of just one or a few selected regions of the entire 16S rRNA gene [16,17,18,19,20], but the obtained results did not provide an exhaustive representation of the biodiversity. In fact, the choice of specific regions of a gene and the corresponding primer pairs, as well as the sequencing methods employed, introduce bias and variability into the results. This can lead to differences in specificity and sensitivity during the analysis, ultimately influencing the overall outcome of the study [21,22,23,24]. Hence, the single-region amplicon-based approach has evolved into two new approaches: the multiplex and the full-length. The multiplex approach tries to mitigate common issues of the single-region approach by simultaneously targeting different hypervariable regions of the selected marker; this approach is expected to overcome the sensitivity limits of a single primer pair, but still relies on short-read sequencing. Conversely, TGS technologies enable the full-length approach, which covers the whole target gene, by using long-read sequencing. Also, a faster library set up and running times, accuracy, and cost-efficiency are TGS advantages. These improvements, together with the development of high-resolution computational methods, have made the full-length approach highly competitive compared to short-read sequencing methods. However, long-read sequencing studies still face limitations. These include higher sequencing error rates, systematic errors, and a lack of mature bioinformatic resources for interpreting the data [25,26,27].

This study compares three different amplicon-based sequencing methods, represented by (a) the single-region approach, testing three different hypervariable regions within the 16S rDNA, V3V4 [28,29], V5V6 [30], and V4 [31,32]; (b) the multiplex approach, and (c) the full-length approach, adopting second- and third-generation sequencing platforms. The aim of this work was to identify the best and the most efficient approach in terms of the processing time, contamination risks, sequencing quality, downstream analysis, high coverage and resolution at the species level, and costs. Firstly, we used a prokaryotic mock community with a known composition, and then the results were validated with biological samples from a mouse model of intestinal inflammation.

Considering the broad impact of microbiome research across various fields that often requires the processing of many samples, our goal was to enhance the efficiency of microbial characterization and offer practical guidelines for obtaining a fast and cost-effective snapshot of the microbiome.

## 2. Materials and Methods

### 2.1. Samples

The commercial mock microbial community, ATCC^®^ 20 Strain Even Mix Genomic Material (MSA-1002^TM^, ATCC^®^, Manassas, VA, USA, https://www.atcc.org/products/msa-1002, accessed on 1 March 2021), composed of a mix of genomic DNA belonging to 20 fully sequenced, characterized, and authenticated ATCC Genuine Cultures (5% for each strain), was used as a benchmark for testing the three different amplicon-based sequencing approaches. The bacterial content of this mock microbial community is described in Appendix A.

The same three approaches were tested on real biological samples: 78 samples (n. 56 feces and n. 22 intestinal content) from an in vivo experimental mouse model of intestinal inflammation. The DNAs were extracted with DNeasy Power Soil Pro kit (Qiagen, Germantown, MD, USA) following the manufacturer’s protocol. The DNAs were provided by the company Postbiotica s.r.l. [29]. 

### 2.2. 16S rRNA Gene Amplicon-Based Library Preparation and Sequencing

Three different 16S rRNA gene amplicon-based sequencing methods were applied to the mock community and to the DNA extracted from feces/intestinal content.

All the sample library preparations for Illumina NGS, according to the single-region and multiplex approaches, were performed, starting from 1 ng of DNA. The QIAseq 16S/ITS Region Panels kit (QIAGEN) was used for targeting the V3V4 hypervariable regions of the bacterial 16S rRNA gene, according to the manufacturer’s instructions. The V5V6 and V4 hypervariable regions of the 16S rDNA were amplified using two overhang primer pairs, BV5/AV6 and 515F/806R, respectively, using the Phusion High-Fidelity DNA Polymerase and following the protocols described in Manzari et al. [33]. 

For the multiplex approach, the Swift amplicon^®^ 16S+ITS Panel Kit (Swift Biosciences, Inc., Ann Arbor, MI, USA) was used, following the manufacturer’s instructions, by applying 22 amplification cycles in the first PCR step and by introducing a purification step of the libraries with AMPure XP beads (0.8×, *v*/*v*) (Beckman Coulter, Inc., Brea, CA, USA) at the end of the protocol. Each NGS library was quality checked through TapeStation 4200 (Agilent Technologies, Santa Clara, CA, USA) or 1.2% agarose gel electrophoresis and quantified using fluorometric assay (Qubit dsDNA HS assay kit, Thermo Fisher Scientific, Waltham, MA, USA). For each different method, the libraries were pooled at equimolar concentrations and sequenced on the MiSeq Illumina platform. The 2 × 300 bp paired-end sequencing strategy (MiSeq Reagent Kit v3, 600-cycle) was used for the V3V4 sequencing on the MiSeq platform. Conversely, the 2 × 250 bp paired-end sequencing strategy (MiSeq Reagent Kit v2, 500-cycle) was used for the V5V6, V4, and multiplex sequencing approaches. 

Sample library preparation for PacBio TGS was performed, starting from 2 ng of total DNA. The 27F/1492R universal primer pair [34] was used to amplify the full-length 16S rRNA gene (corresponding to the region from V1 to V9), with both primers tailed with sample-specific PacBio barcode sequences for multiplexed sequencing (for the barcode sequences, see the Appendix of the PacBio protocol “Amplification of Full-Length 16S Gene with Barcoded Primers for multiplexed SMRTbell^®^ Library Preparation and Sequencing”—Version 04—January 2021). Amplifications were performed using a 25 µL mixture containing 2 ng of DNA, 1× Buffer HF, 0.2 mM dNTPs, 0.375 µM of each primer (Fwd-Rev), and 1U/μL of Phusion High-Fidelity DNA Polymerase. The cycling parameters for amplification of the full-length 16S rRNA gene were standardized as follows: initial denaturation at 98 °C for 30 s, followed by 10 cycles of denaturation at 98 °C for 10 s, annealing at 58 °C for 30 s, extension at 72 °C for 1 min, and subsequently, 15 cycles of denaturation at 98 °C for 10 s, annealing at 62 °C for 30 s, extension at 72 °C for 1 min, with a final extension step of 7 min at 72 °C. The PCR products (~1.5 kb long) were quality checked by 1.2% agarose gel electrophoresis and quantified using Qubit 1× dsDNA HS Assay kit. Each barcoded PCR reaction product was pooled at equimolar concentrations. Then, the SMRTbell library construction was performed according to the manufacturer’s instructions of the “Amplification of Full-Length 16S Gene with Barcoded Primers for multiplexed SMRTbell^®^ Library Preparation and Sequencing” (Version 04—January 2021), starting from the step “Pooling Barcoded Amplicons”. The SMRTbell^®^ Express Template Prep Kit 2.0, the Binding Kit 2.1, the Sequel II Sequencing Kit 2.0, and a single SMRT^®^ Cell 8M (Pacific Biosciences, Menlo Park, CA, USA) were used for library preparation and sequencing on the PacBio Sequel II System.

All PCR reactions were performed in the presence of a negative control (Molecular Biology Grade Water, RNase/DNase-free water), to exclude contamination during library preparation steps. The negative controls were evaluated qualitatively and quantitatively, just like the other samples. As the analyses confirmed the absence of contaminations, these controls were not sequenced.

### 2.3. Bioinformatic Data Analysis

#### 2.3.1. Single-Region Data Analysis

Illumina adapters and PCR primers were removed from raw reads by applying cutadapt (version 4.0, parameters for adapter trimming -a CTGTCTCTTATACACATCT -A CTGTCTCTTATACACATCT, parameters for primer trimming -g “FORWARDPRIMER;e = 0.4” -G “REVERSEPRIMER;e = 0.4” -m 100 --discard-untrimmed) [35]. The retained paired-end reads were analyzed by DADA2 (version 1.22, the following parameters were used for filtering: maxN = 0, maxEE = c(5,5), truncQ = 0, rm.phix = TRUE, compress = TRUE; for error model inference MAX_CONSIST = 20, randomize = T; for paired-end merging minOverlap = 8, maxMismatch = 2; and chimera removal method = “pooled”) [36] in R4.1.3 to reduce the amplification and sequencing noise and obtain ASVs (amplicon sequence variants). ASVs were taxonomically annotated by using the QIIME2classify-sklearn plugin [14] (version qiime2-2022.2, default parameters) and the release 138 NR 99 of the SILVA database [37]. 

#### 2.3.2. Multiplex Data Analysis 

The bioinformatic analysis was performed by using the Swift workflow available on GitHub (https://github.com/swiftbiosciences/16S-SNAPP-py3, accessed on 29 September 2021). Initially, raw reads were treated for primer trimming by cutadapt (version 4.0) [35]. Untrimmed reads (i.e., reads in which primer sequences were not found) were discarded. Next, the retained reads were denoised in ASVs by DADA2 (version 1.22) [36]. We refer to these ASVs as preliminary (pASVs). The obtained pASVs were initially classified through the RDP classifier [38]. Then, iterative steps of mapping the obtained ASVs against a reference collection of whole 16S rRNA sequences (i.e., SILVA release 138 NR 99; [37]), by using the nucleotide blast (blastn version 2.9.0+), were performed to identify the closest reference sequence with the highest probability (in terms of similarity and coverage) to have generated the observed pASVs. The workflow results were available in three tabular (tsv format) files: (i) a feature table summarizing the number of times a 16S rRNA sequence is observed in each sample, (ii) a lineage table containing the observed taxa counts in each sample based on the RDP classifier data, and (iii) a taxonomy table containing the taxonomic classification for each identified 16S rRNA sequence based on the RDP classifier data. In order to obtain comparable taxonomic classification with the other methods, the selected 16S rRNA full-length sequences were reconstructed by using a Python script and taxonomically classified by using the QIIME2classify-sklearn plugin [14] (version qiime2-2022.2, default parameters) and the release 138 NR 99 of the SILVA database [37].

#### 2.3.3. Full-Length Data Analysis

PacBio HiFi (high-fidelity circular consensus sequences) were analyzed by using a workflow relying on DADA2 [36] (version 1.22) in R4.1.3 and its optimization for these kinds of data. Initially, it starts by trimming reads using the primer sequences (primer.fwd = “AGRGTTYGATYMTGGCTCAG”, primer.rev = dada2:::rc(“RGYTACCTTGTTACGACTT”), orient = TRUE). Moreover, since PacBio adapters are added through ligation, it verifies that all the reads have the same orientation (Fwd-Rev). Following data filtering to remove noisy reads (minLen = 1000, maxLen = 1600, maxN = 0, rm.phix = FALSE, maxEE = 2) [39], the error model was estimated by using a function specifically designed for PacBio data (BAND_SIZE = 32, errorEstimationFunction = dada2:::PacBioErrfun). The inferred error model was used to denoise the reads and for ASV estimation. Finally, ASVs were chimera checked (method = “pooled”, minFoldParentOverAbundance = 3.5). The retained ASVs were taxonomically annotated as per the previous approaches, i.e., by using the QIIME2classify-sklearn plugin [14] (version qiime2.2022-2, default parameters) and the release 138 NR 99 of the SILVA database [37].

#### 2.3.4. Statistical Analysis 

The analyzed mock community consists of an equal genome mass of each of the 20 bacterial species. Therefore, considering the variability in genome size and in 16S rDNA copy number between the mock species (Appendix A), we needed to estimate the number of expected 16S rDNA copies per genome of the mock to infer the expected 16S rDNA relative abundances. Estimates were made as follows.

Initially, we defined the mass of each genome. Since the average weight of a base pair in dsDNA is 607.4 g/mol, we calculated the genome molecular weight for each species “i” of the mock, i = 1, 20, (GMW_i_) as follows: GMW_i_ [g/mol] = GenomeLength_i_ × 607.4 (g/mol)(1)

Next, the genome mass in ng (nGM) was inferred:(2)$${\mathrm{nGM}}_{i}\;[\mathrm{ng}]=\frac{\mathrm{GMWi}\;\left(g/\mathrm{mol}\right)}{\left(6.022\times 10^{23}\right)\left({\mathrm{mol}}^{-1}\right)\;}\times 10^{9}$$

Then, we estimated the genome copy number (GCN) into a specific mass. Considering the concentration of the purchase mock DNA mix was 2.6 ng/uL in a volume equal to 50 uL, we calculated the mock mass per species (Mock Mass) and then the genome copy number for each species i (GCN_i_): (3)$$\mathrm{Mock}\;\mathrm{Mass}\;[\mathrm{ng}]=\frac{2.6\;\left(\mathrm{ng}/\mathrm{uL}\right)\;\times 50\left(\mathrm{uL}\right)}{\left(20\right)\;}=6.5\;\mathrm{ng}$$
(4)$${\mathrm{GCN}}_{i}=\frac{\mathrm{Mock}\;\mathrm{Mass}\;\left(\mathrm{ng}\right)}{\mathrm{nGMi}\;\;\left(\mathrm{ng}\right)}$$

Finally, knowing the GCN_i_, we inferred the number of expected 16S rDNA copies per each species in the mock:Expected 16S copies = GCN_i_ × 16S Genome copies(5)

The per species expected 16S copies were used to infer the expected relative abundances, simply by dividing each single species 16S gene expected copies to the total one (Appendix A). The correlation between the expected and observed 16S rRNA relative abundances (%) for each bacterium of the benchmark was investigated for the three sequencing methods, via the linear regression model and Pearson correlation coefficient. 

Raw reads retained following primer trimming were mapped on the mock species reference genomes by using minimap2 (version 2.17, short reads: --eqx -t 10 --MD -ax sr; long reads: --eqx -t 10 --MD -ax map-pb) [40]. The obtained alignment was parsed by using a Python script relying on pysam module (version 0.21), a wrapper around the samtools [41] through the fetch, and pileup functions, allowing us to count the number of reads mapping on 16S rRNA genes and their coverage, respectively. Considering the high similarity in 16S rRNA genes belonging to co-generic species, only primary alignments were considered. For paired-end data, an additional control was performed in order to retain only read pairs mapping on the same 16S rRNA genes or on identical copies (i.e., forward and reverse reads mapped on different but identical copies of the 16S rRNA genes in the same genome). The discarded pairs were labeled as ambiguously mapped on target regions. The same procedure was also applied to pASVs inferred with the Swift workflow. Mapped reads were stratified in three main groups: mapping on target 16S rDNA genes, out of target, and unmapped.

The inferred single-region and full-length ASV, and the selected 16s rRNA gene for the multiplex approach, were aligned against the mock genomes by using nucleotide blast [42,43,44] (v 2.12.0+, identity percentage ≥ 97% and query-coverage ≥ 90%). Next, taking into account the annotation available per each mock genome and the obtained taxonomic classifications, we built a confusion matrix at species and genus levels per each tested approach as follows: ASV mapping on genomic regions annotated to contain 16S rRNA genes and correctly classified were labeled as true positive (TP); those mapping on genomic regions annotated for 16S rRNA genes and not correctly classified were false positive (FP); ASVs mapping on genomic regions not containing 16S rRNA genes were true negative (TN); and ASVs mapping on genomic regions annotated to contain 16S rRNA genes but not classified at all were false negative (FN). The precision, accuracy, and recall were measured by using the obtained confusion matrices. Finally, the receiver operating characteristics (ROC) curves and area under the curve (AUC) score were obtained by using sklearn [45].

For the murine biological samples analysis, contaminant ASVs were identified and removed by using the R packages decontam (version 1.16) [46] by using the frequency method that relies on the identification of ASVs whose abundance is inversely related to DNA concentration. Moreover, ASVs labeled as plastid or mitochondrial were removed from subsequent analysis. The R packages phyloseq (version 1.40) [47] and vegan (version 2.6.4) [48] were used to measure α and β diversity. ASV counts were normalized by using rarefaction to perform α diversity inference (i.e., intra-sample diversity) and then, the inverse Simpson and Pielou’s evenness indexes were calculated. Statistical differences in α diversity indexes were measured by using the paired Student’s *t*-test (*p* < 0.05 was considered as statistically significant). According to Gloor et al. [49], and taking into account the compositional nature of metabarcoding data, the β-diversity (i.e., inter-sample diversity) was measured by transforming the data through CLR (centered log ratio) and measuring inter-sample distances with the Aitchison distance. To simplify data interpretation, principal coordinates analysis (pCoA) was applied to reduce data dimensionality. The permutational analysis of variance was measured to infer the explained variability in β diversity data by applying 999 permutations. 

The abundance comparisons among sequencing methods for taxa were performed by ANCOM-BC (version 1.6.4) analysis [50] in a pairwise manner. ANCOM-BC tests were performed by adjusting the obtained *p*-value by the Benjamini–Hochberg false discovery rate correction method. Log fold change > |1| and adjust *p*-value < 0.05 were considered for statistical significance. 

## 3. Results

### 3.1. Mock Benchmark Analysis

A mock community was used as an experimental and bioinformatic benchmark in order to evaluate the efficiency of the three amplicon-based sequencing methods, i.e., the single-region, multiplex, and full-length approaches, to provide an accurate microbiome characterization. The mock microbiome community that we analyzed contained bacteria that were both frequent and rare in the human microbiome, under eubiosis and dysbiosis conditions. It covers the phyla Bacteroidota (Bacteroides vulgatus, Porphyromonas gingivalis), Actinobacteriota (Bifidobacterium adolescentis, Cutibacterium acnes, Schaalia odontolytica), Firmicutes (Bacillus pacificus, Clostridium beijerinckii, Enterococcus faecalis, Lactobacillus gasseri, Staphylococcus aureus, Staphylococcus epidermidis, Streptococcus agalactiae, Streptococcus mutans), Proteobacteria (Acinetobacter baumannii, Escherichia coli, Helicobacter pylori, Neisseria meningitidis, Pseudomonas paraeruginosa, Cereibacter sphaeroides), and Deinococcota (Deinococcus radiodurans), for a total of 18 genera and 20 bacterial species (Appendix A). For each method, the amplicon libraries were sequenced by a specific platform and the results were compared at genus and species level. As for the genus-level comparisons (Table 1), the V3V4, V4, and full-length approaches gave the best results since they identified the 18 expected genera. Moreover, a strong linear correlation (Pearson’s r = 0.76 for V3V4, r = 0.70 for V4, and r = 0.71 for full-length; *p*-value < 0.001) was observed between the expected and observed 16S rRNA relative abundances identified at the genus level (Figure 1). Otherwise, the V5V6 single-region and the multiplex approach identified only 10/18 and 10/18 genera, respectively, without a significant statistical correlation between the expected and observed 16S rRNA abundances, as shown in Figure 1. 

Moving to the species-level analysis (Table 2), in the single-region approach, 4 species, 12 species, and 9 species of the mock community were identified by the V5V6, the V3V4, and the V4 targets, respectively. Additionally, only for the V3V4 target was a weak correlation between the expected and observed 16S relative abundances evaluated (Pearson’s r = 0.26; *p*-value = 0.251), even if no statistical significance was revealed (Figure 2). The multiplex approach detected only four species, and the 16S rDNA-associated counts were a lot lower than expected (Figure 2, Appendix A). The full-length approach detected 3 more species than the V3V4 single-region approach, represented by *Clostridium beijerinckii*, *Enterococcus faecalis,* and *Pseudomonas aeruginosa*, for a total of 15 observed species. The observed counts by the full-length approach show a weak linear correlation (Pearson’s r = 0.28; *p*-value = 0.223) (Figure 2). No methods identified the two species of the genus *Staphylococcus* (*S. aureus* and *S. epidermidis*), except for *S. epidermidis* which was identified only by the V5V6 region. On the contrary, only the V3V4, V4, and full-length approaches were able to distinguish the species *Streptococcus agalactiae* and *Streptococcus mutans*.

To further investigate the failure in the classification of some taxa and identify the eventual limiting step of the entire workflow, we initially mapped the raw PE reads, retained following the primer trimming, on the mock species reference genomes. For short-read mapping, we considered only PE reads mapping on the same 16S rRNA genes or on an identical copy. PE reads mapping on different non-identical copies of the SSU genes were considered ambiguous and not considered for subsequent analysis. Finally, we also considered PE reads mapped in the genomic region out of those annotated to contain 16S rRNA genes and unmapped ones. The largest amount of read mapping on 16S rRNA genes was observed in full-length (99.98%) and V4 (98.76%) (Appendix A). V5V6 obtained the lowest amount of PE reads mapping unambiguously on SSU genes (15.04%) and the highest rate of ambiguous ones (78.40%) (Appendix A). Reads mapping on unexpected regions were observed only in V4 (0.03%) and V3V4 (0.001%) (Appendix A). Finally, the topmost unmapped rates were observed in multiplex (13.14%) and V5V6 (6.57%) (Appendix A). In Appendix A and Appendix A are shown the relative abundances of reads mapping on 16S rRNA genes per each species represented in the mock community. It was only via the multiplex approach that we did not find reads mapping to all the mock species (in particular *D. radiodurans* and *A. baumannii*). A relevant positive Pearson correlation among the expected and observed relative abundances was observed for V3V4 (Pearson’s r = 0.61; *p*-value = 0.004), V4 (Pearson’s r = 0.60; *p*-value = 0.005), V5V6 (Pearson’s r = 0.51; *p*-value = 0.022), and full-length (Pearson’s r = 0.54; *p*-value = 0.015). Moreover, considering the Swift bioinformatic workflow implements a preliminary denoising step before the selection of 16S rRNA sequences from the reference collection, we decided to evaluate pASVs and map them on the reference mock genomes. We observed these pASVs mapped on 12 out of 20 species. Finally, we have also evaluated and compared the observed 16S rRNA gene coverage in multiplex raw reads and pASVs (Appendix A). Regarding the raw reads, in 6 out of 18 species (namely *B. adolescentis*, *C. beijerinckii*, *E. fecalis*, *E. coli*, *L. gasseri,* and *B. vulgatus*), the expected pattern, covering the whole 16S rRNA gene, was revealed. The same pattern was also observed at the pASV level, with the exception of *C. beijerinckii*. Furthermore, regardless of the analyzed species, the coverage pattern was uneven. Regarding the mapped pASVs, we overall observed the same coverage pattern as seen at the raw reads level in *B. pacificus*, *N. meningitidis*, *R. sphaeroides*, *S. aureus*, *S. agalactiae*, *S. epidermidis,* and *S. mutant*. 

*S. odontolytica* had a different trend, showing a profile of pattern coverage higher than the one observed in the raw reads.

Subsequently, the ASVs derived from each single-region and full-length approach and 16S rRNA full-length sequences selected by the multiplex approach were mapped to the available reference genomes of the mock species (BLAST analysis with query coverage ≥ 90% and identity ≥ 97%) (Appendix A). For the V3V4 and full-length methods, all of the generated ASVs (20 out of 20) were aligned to the reference genomes of the mock microbiome. For the V5V6 and V4 methods, 19 ASVs were aligned, and only 6 ASVs were identified using the multiplex method. The analysis also highlighted the presence of background noise, including false positives, associated with these methods (Appendix A). The precision, accuracy, and recall (sensitivity) values for each method are provided in Table 3 and Appendix A.

### 3.2. Validation on Real Microbiome Samples

Taking into account the results of the benchmark study, the three different 16S rDNA amplicon-based approaches (i.e., the V3V4 single-region approach, see Discussion, the multiplex, and the full-length methods) were used to analyze 78 real samples obtained from the feces/intestinal content of a mouse model of intestinal inflammation.

#### 3.2.1. Sequencing Output and Data Processed 

Three separate sequencing runs were performed, generating the output shown in Table 4. About 28 million reads (mean/sample = 180,769 ± 39,538) and about 13.5 million reads (mean/sample = 146,153 ± 57,231) were generated across all samples by Illumina MiSeq sequencing while about 2.5 million HiFi reads (mean/sample = 28,263 ± 12,487) were generated by PacBio sequencing.

Following the trimming, merging, and denoising procedures, about 88.6%, 76.0%, and 53.1% of the initial sequences were retained for the V3V4, multiplex, and full-length data, respectively. Overall, 1150 ASVs and 1715 ASVs for V3V4 and full-length, respectively, were retained without contaminants, chloroplast, and mitochondrial sequences. Regarding the multiplex approach, we retained 52,534 full-length sequences.

For ecological metrics inference, data were normalized by rarefaction to the number of 100,000 sequences for the V3V4 data, 52,000 sequences for the multiplex data, and 7427 sequences for the full-length data (Figure 3). To include all samples in the analysis, the minimum count among the sequences of interest was used as the base value for each approach. 

#### 3.2.2. α and β Diversity Analysis

The α (i.e., intra-sample) diversity was investigated by using the inverse Simpson and Pielou’s evenness indices, starting from normalized ASV counts. Statistically relevant differences (*p*-value < 0.05) in both metrics were observed by using the paired *t*-test between the sequencing method sets (Figure 4). Overall, the multiplex method was characterized by the highest diversity according to the inverse Simpson index (Figure 4a), whereas Pielou’s evenness index resulted in the highest in the full-length method (Figure 4b), with all comparisons resulting in being statistically significant. In both the metrics, the diversity values obtained via the V3V4 approach were contained in the range of values of the full-length approach (Figure 4a,b). 

Additionally, α diversity indices were computed using genus-level counts for each sequencing approach (Figure 5). The results of this analysis indicated that the V3V4 and full-length approaches captured similar levels of biodiversity, with no statistically significant differences in pairwise comparisons. However, both of these methods yielded statistically higher α diversity compared to the multiplex approach (*p*-value < 0.05).

The β (i.e., inter-sample) diversity was measured by transforming the data through CLR (centered log ratio) and measuring inter-sample distances by applying the Aitchison distance to the unified taxonomic counts. The PCoA plot for the phylum-level data (Figure 6a) shows a clear separation between the multiplex data and V3V4/full-length data along the first component (PCoA1, 72.46%). The same trend of separation was observed for the family- and genus-level data too (Figure 6b,c) (PCoA1 68.4% and PCoA1 57.6%, respectively). On the contrary, from the phylum- to genus-level data (Figure 6a,c), the second component does not provide a sufficient variability for the separation between the three methods (PCoA2, 8.7%; PCoA2, 4.81%; PCoA2, 6.21%, respectively). Overall, the V3V4 and full-length data cluster together along the first component. Interestingly, at the species level (Figure 6d), the three methods separated well along both components (PCoA1 48.95% and PCoA2 13.36%). We used Permanova analysis with the Aitchison distance to compare the variability of the three methods. At the species level, the methodological approach of MPX differed greatly from V3V4 and FL, explaining 71.35% and 59.64% of the variation, respectively. In contrast, V3V4 and FL only differed by 27.65% (Table 5). 

#### 3.2.3. Taxonomic Characterization of the Microbiome 

The observed ASVs identified by each sequencing method were taxonomically annotated against the SILVA database. Venn diagrams were used to depict unique and common taxa for each method at different taxonomic levels, from phylum to species. We only included taxa with a relative abundance of 1% or more. As shown in Figure 7, the three methods exhibited a full overlap of the community composition only at the phylum level. The number of unique and shared taxa varied at lower taxonomic levels. Interestingly, at the species level, the full-length approach identified 14 exclusive taxa, more than the V3V4 and multiplex methods, which identified 10 and 5 exclusive taxa, respectively.

Taxonomic assignments and relative abundances at the phylum, family, genus, and species levels are shown as donut charts (Figure 8). Only taxa with a relative abundance (RA) equal to or higher than 1% were plotted; otherwise, they were collapsed into “Others”. Pairwise comparisons between each sequencing method were performed using ANCOM-BC (Appendix A). The taxa identified by the statistical analysis at the phylum and species levels, with lfc > |1| and adjust *p*-value < 0.001, were plotted in Figure 9.

Bacteroidota, Firmicutes, Verrucomicrobiota, Actinobacteriota, and Proteobacteria were the principal phyla identified by the three methods. Bacteroidota (50.0% ± 9.84% in V3V4, 26.0% ± 6.13% in MPX, 41.1% ± 8.20% in FL) and Firmicutes (40.4% ± 11.59% in V3V4, 45.2% ± 11.12% in MPX, 47.1% ± 11.90% in FL) were assigned as dominant phyla, followed by the others. However, the microbiome composition determined by each sequencing method was statistically different at lower taxonomic levels. In particular, the microbiome characterized by the V3V4 and full-length methods was mostly represented by the genera Muribaculaceae (p. Bacteroidota; f. Muribaculaceae); Prevotellaceae_UCG-001 (p. Bacteroidota; f. Prevotellaceae); Lactobacillus (p. Firmicutes; f. Lactobacillaceae) with equally distributed species *L. murinus* and *L. reuteri*; Lachnospiraceae_UCG-001; Lachnospiraceae_NK4A136_group; Lachnoclostridium (p. Firmicutes, f. Lachnospiraceae); Ruminoccocus (p. Firmicutes; f. Ruminococcaceae); Parasutterella (p. Proteobacteria; f. Sutterellaceae), and Akkermasia (p. Verrucomicrobiota; f. Akkermansiaceae) with the species *A. muciniphila*. All these taxa resulted in being statistically more abundant in the two methods mentioned above than in the multiplex method (lfc > |1|, *p*-value < 0.05). Moreover, only the genus Faecalibaculum (p. Firmicutes; f. Erysipelotrichaceae) was statistically more abundant in the full-length data, whereas the species *L. johnsonii* (g. Lactobacillus) and *B. pseudolongum* (g. Bifidobacterium) were exclusively identified by the full-length. In contrast, the multiplex method characterized a microbiome statistically composed of the genera Muribaculum (p. Bacteroidota; f. Muribaculaceae) with the species *M. intestinale*, Alistipes (p. Bacteroidota; f. Rikenellaceae), Bacteroides (p. Bacteroidota, f. Bacteroidaceae), Tepidibacter (p. Firmicutes; f. Peptostreptococcaceae), Bacillus (p. Firmicutes; f. Bacillaceae) with an incorrectly classified species *L. vaginalis*, Bifidobacterium (p. Actinobacteriota; f. Bifidobacteriaceae), Amphiplicatus (p. Proteobacteria; f. Parvularculaceae), and Pigmentiphaga (p. Proteobacteria; f. Alcaligenaceae). Unusually, the only species of the genera Bacteroides and Bifidobacterium with RA > 1% were *B. acidifaciens, B. choerinum,* and *B. pseudolongum*; these were not identified by the multiplex method, but by the other methods. Finally, from the phylum to family levels, no unclassified ASVs were observed for the full-length method. Finally, the full-length method did not have any unclassified ASVs from the phylum to family levels. However, the rate of unclassified ASVs increased slightly at the genus level, but it was still the lowest among the methods. On the other hand, the multiplex method had the highest rate of unclassified ASVs already at the phylum level.

## 4. Discussion

To study the prokaryotic microbiome, the most widely used approach is amplicon-based sequencing that focuses on one or a few regions of the 16S rRNA gene. This approach has been progressively improved following technological advances and the demands of modern research [11]. Currently, two amplicon-based approaches represent a possible alternative: the multiplex and the full-length methods [34,51,52]. The first represents a compromise that targets different regions of the marker gene relying on short-read sequencing, whereas the second takes advantage of long-read sequencing to cover the whole length of the gene. 

NGS has lower analytical costs and uses established bioinformatic pipelines and databases for downstream analysis, but nowadays, TGS has become more competitive thanks to cost reduction and the development of efficient analysis methods [22,24]. In both cases, the experimental workflows lead to sample handling improvements by reducing execution times and contamination risk. The full-length method is especially advantageous because it handles pooled samples from the beginning of library preparation. Therefore, the main challenge remains to identify the amplicon-based sequencing approach that can best reliably characterize the microbiome up to lower taxonomic levels, such as the species level. So, the present study investigates the effectiveness of three different experimental approaches, the single-region and multiplex using the NGS platform Illumina MiSeq, and the full-length using the TGS platform PacBio. We first performed the microbial characterization on a prokaryotic mock community as a benchmark for both the experimental and bioinformatics steps. Then, we applied the same methods to a cohort of complex murine samples to validate the results. 

A benchmarking study is crucial to understand the relative performance of taxonomic profiling methods for different purposes, providing a ”ground truth” to which results can be compared [53,54]. The prokaryotic mock microbiome considered is composed of 20 bacterial species, representative of 18 genera. The results show the different sensitivity of each hypervariable region of the 16S rRNA gene in microbiome profiling, supporting that the choice of a region can affect the results [21,22,55] and lead to intrinsic analytical limitations due to the sequence features of the reference collection. Indeed, it is evident, already at the genus level, that the taxa identified by the single-region approach vary depending on the hypervariable region chosen (V3V4, V5V6, and V4) (Table 1). Nevertheless, the full-length approach identifies with high fidelity 18/18 genera, the same as the V3V4 and V4 methods. It is worthy to note that for both *Clostridium beijerinckii* and *Schaalia odontolytica,* the genera associated with the reference taxonomy were *Clostridium senso stricto 1* and *Actinomyces*, respectively. This represents a classification issue and classifier performances are greatly affected by the reference database [56] and this becomes even more evident at the species level. The full-length approach identifies the highest number of species (15/20), overcoming the discrimination power of the others (Table 2). Despite the sequenced ASVs matching with the available reference sequences of the mock community, some species are not classified. Indeed, the SILVA database [57], with more reference sequences than other 16S rRNA gene databases such as GreenGenes and RDP [58,59], includes a considerable number of taxonomies that do not have the resolution to the species level. Missing a species in the database can result in misclassification [60], limiting the classifiers’ performance. Therefore, the results at the species level in the mock community are not related to the experimental protocol but to a classification issue [56]. In addition, we detected unexpected ASVs not matching with mock reference genome sequences (Appendix A). These data support the use of ≥1% relative abundance as a suitable threshold able to eliminate the background noise in taxonomic analysis in the case of single-region analysis. This threshold may become ≤0.1% in the case of FL analysis, thus remarkably improving detection sensitivity and specificity.

Despite the multiplex approach being characterized by a higher gene coverage compared to the single-region approach, its resolution at the genus level was comparable to the V5V6 target, but its performance was limited compared to the other approaches (Table 1 and Table 2). This limitation could be related to both the experimental procedure and the different steps of the specific bioinformatic workflow. Indeed, when mapping the raw reads on reference mock genomes, we found that only 18 out of 20 species were detected and this supports the idea of a possible issue in multiplexed PCR efficiency. Regarding the bioinformatic approach, it involves the selection of 16S full-length sequences from a reference collection that have a higher probability to be observed in the analyzed dataset. This selection relies on the observed pASVs and in our analysis, we demonstrated the impact of the denoising step. Indeed, considering pASVs, only 12/20 species were detected and probably the implemented approach fails in discriminating among noise and real sequence variability and this results in an aggregation of sequences belonging to different taxa. A remarkable example is *S. odontolytica* for which we observed more sequences associated with pASVs compared to raw ones. These results support the thesis that issues in both experimental and bioinformatic steps limit the reliability of the multiplex approach. In our study, the mock community, which is usually used as an internal control to monitor the entire sequencing workflow, was sequenced and analyzed with real intestinal biological samples. This might influence the multiplex approach by affecting the capacity of DADA2 to distinguish real biodiversity from noise. Indeed, in our results, we clearly show how, following the denoising step, we were unable to map pASVs to eight species. Once again, it is crucial to note that denoising is preliminary to 16S full-length sequence selection, and the loss of data introduces biases in the subsequent analytical steps. Furthermore, the shown data represent cumulative results obtained following an experimental and bioinformatic workflow. Consequently, it is difficult to define whether we are observing, for instance, an issue related to primer amplification bias or to bioinformatic analysis. 

The mock benchmark has enabled the optimization of amplicon-based sequencing workflows, but the microbiome communities of biological samples are much more complex than a mock community [24]. So, the previous results have been validated using a community of 78 fecal samples deriving from a specific stratified sampling. Considering the higher resolution achieved in the preliminary benchmark analysis by V3V4 compared to the other single regions, for the single-region approach, just the V3V4 region has been chosen as the target for the validation. 

As explained by the rarefaction curves (Figure 3), both the full-length and the V3V4 approach can reliably estimate all the community diversity, instead of the unreached plateau for the multiplex data. These data are confirmed by Pielou’s index calculation which shows a greater evenness of the community for the full-length approach, which is therefore able to capture a higher biodiversity [61]. At the same time, the higher value of the inverse Simpson index recorded for the multiplex approach seems to not be associated with a real greater biodiversity but probably with the overestimation of ASVs (https://github.com/swiftbiosciences/16S-SNAPP-py3, accessed on 1 January 2020), as demonstrated by the related low Pielou’s index inference (Figure 4). This discrepancy is additionally confirmed by the α diversity analysis performed with both indexes at the genus level (Figure 5), highlighting the lower biodiversity captured by the multiplex method compared to the other methods.

The ability of each approach to reliably detect the specific biodiversity is also revealed by the β diversity analysis. The biodiversity measured by the full-length and V3V4 approaches overlaps at the higher taxonomic levels and shows a good separation only at the species level (Figure 6). On the contrary, the multiplex data cluster separately at all the taxonomic levels analyzed. They show a different community composition from the one identified by the other two methods (Figure 6). Focusing on the taxonomical analysis, the three methods identify the same five phyla, represented by Bacteroidota, Firmicutes, Verrucomicrobiota, Actinobacteriota, and Proteobacteria (Figure 7 and Figure 8). In the multiplex data, an increase in the phyla Proteobacteria and Actinobacteriota and a decrease in the phylum Verrucomicrobiota are observed. Moving to lower taxonomic levels, the relationship between the V3V4 and full-length methods becomes evident. In fact, at the family and genus level, all the taxa identified by the full-length method match with those found by the V3V4 method. On the contrary, the multiplex method shared only 9 taxa at the family and 10 taxa at the genus level with the V3V4 and full-length methods. Finally, the highest number of species were identified by the full-length method, of which 14 taxa were exclusive, 29 taxa were shared with the V3V4 method, and 3 taxa were shared with the multiplex method. In particular, the full-length method can classify two exclusive species represented by *Bifidobacterium pseudolongum* and *Lactobacillus johnsonii*, five species common to the V3V4 method as *Akkermansia muciniphila, Bacteroides acidifaciens, Lactobacillus murinus, Lactobacillus reuteri,* and one species (*Muribaculum intestinale*) also identified by the multiplex method. Within the genus *Bifidobacterium*, the full-length and V3V4 methods recognized two different species, represented by *Bifidobacterium pseudolongum* and *Bifidobacterium choerinum*, respectively. In the case of the multiplex, the exclusive species *Lactobacillus vaginalis* is misclassified to the genus *Bacillus*, creating a large variability in the overall composition of the microbial community, compared to the other methods. These data confirm the trend observed in the β diversity analysis and highlight the ability of the full-length method to capture the same microbiome profile identified by the single-region method but at the same time, overcoming the resolution of both the single-region and multiplex methods. 

The last important aspect to consider is the rate of unclassified taxa for each method (Figure 8). The full-length approach does not have any unclassified ASVs up to the family level and maintains the lowest rate at the genus level. The multiplex method, on the other hand, has a higher rate of unclassified taxa, starting from the phylum level. These last results confirm the limits associated with downstream analysis and the pipeline used for analyzing multiple small amplicons and reconstructing the 16S rRNA gene.

## 5. Conclusions

This comparative assessment between the three amplicon-based methods verified the reliability of the single-region-based study. However, it also demonstrated the better performance of the complete target analysis and its higher effectiveness on complex biological communities. In fact, the analysis, supported by a case study, highlighted the greatest discriminating power of the full-length 16S rRNA approach up to the species level. It also benefited from its less laboriousness, lower execution time, and contamination risk, at a similar cost to the standard single-region approach. Hence, the amplification of the whole 16S rRNA gene and the use of TGS demonstrated an improvement, in both experimental and downstream analysis, compared to previous methods. Despite the issues with reference databases [56], this approach was able to identify more species of the known composition benchmark and to exclusively classify *B. pseudolongum* and *L. johnsonii* within the real dataset, as compared to the short-read sequencing approaches. On the other hand, the multiplex method presented remarkable flaws, such as sequencing depth and sampling, inference of ASVs, and 16S reference collection (https://github.com/swiftbiosciences/16S-SNAPP-py3, accessed on 1 January 2020). 

Considering all these factors, this study supports the transition from NGS to TGS for the study of the intestinal microbiome, opening a new frontier in biomedical research to revolutionize the way to act against disease conditions [62]. Further studies may be performed to demonstrate the effectiveness of this approach for samples of a different nature and taxonomic complexity, such as environmental samples.

## Figures and Tables

**Figure 1 genes-14-01567-f001:**
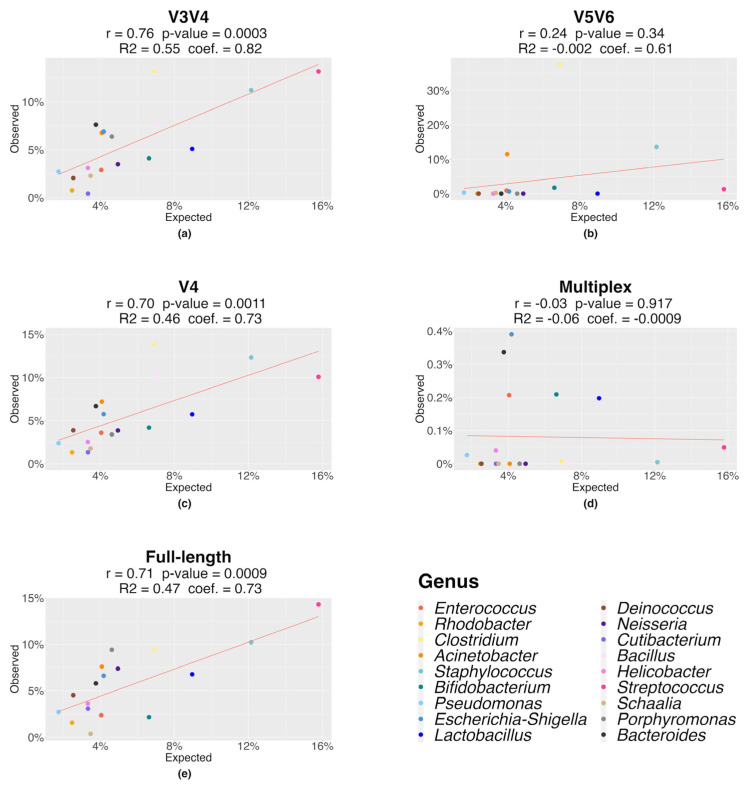
Correlation between the expected and observed 16S rRNA relative abundances (%) at the genus level for each sequencing approach. Correlation for (**a**) the V3V4 region; (**b**) the V5V6 region; (**c**) the V4 region; (**d**) the multiplex approach; (**e**) the full-length approach. For each method, the adjusted R2, linear model coefficient (coef.), Pearson correlation coefficient (r), and *p*-value are shown.

**Figure 2 genes-14-01567-f002:**
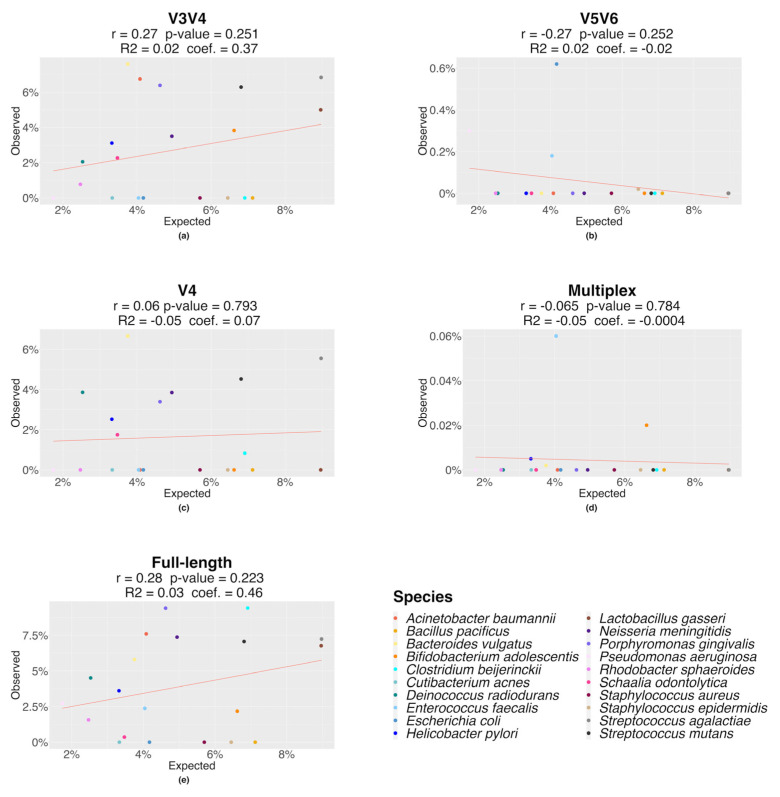
Correlation between the expected and observed 16S rRNA relative abundances (%) at the species level for each sequencing approach. (**a**) Correlation for the V3V4 region; (**b**) correlation for the V5V6 region; (**c**) correlation for the V4 region; (**d**) correlation for the multiplex approach; (**e**) correlation for the full-length approach. For each method, the adjusted R2, linear model coefficient (coef.), Pearson correlation coefficient (r), and *p*-value are shown.

**Figure 3 genes-14-01567-f003:**
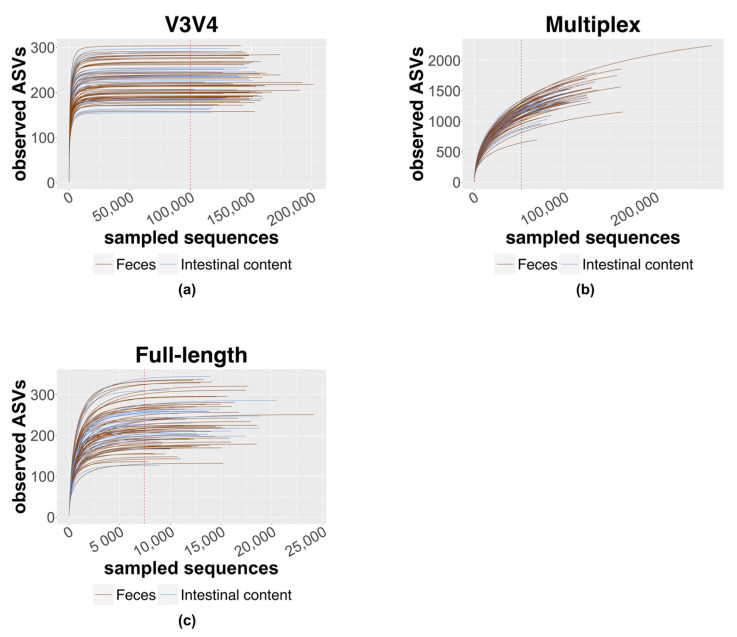
Rarefaction curves of sequencing data. The figure shows the rarefaction curves of (**a**) V3V4 sequencing data with a threshold of 100,000 sequences; (**b**) multiplex sequencing data with a threshold of 52,000 sequences; (**c**) full-length sequencing data with a threshold of 7427 sequences. In brown are shown the rarefaction curves of feces samples, whereas in blue are shown the rarefaction curves of intestinal content samples.

**Figure 4 genes-14-01567-f004:**
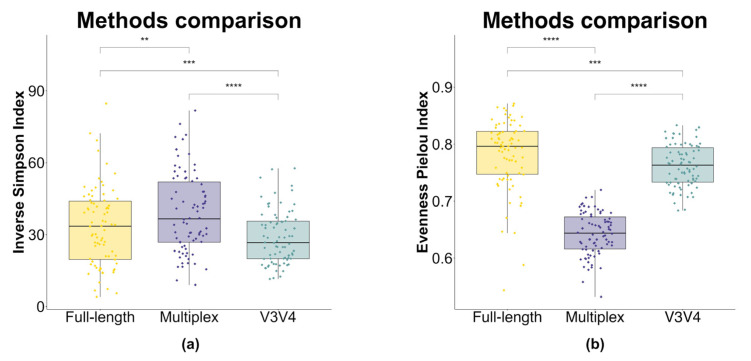
Box plot of α diversity indices calculated for each sequencing method. (**a**) α diversity measured using the inverse Simpson index. (**b**) α diversity measured using Pielou’s evenness index. α diversity scores were calculated by using rarefied ASV counts for each approach. Box plots and points represent the overall data distribution and single samples, respectively. Yellow: full-length approach; violet: multiplex approach; water-green: V3V4 approach. The group means comparison was performed by using the paired Student’s *t*-test (“**”: *p*-value < 0.01; “***”: *p*-value < 0.001; “****”: *p*-value < 0.0001).

**Figure 5 genes-14-01567-f005:**
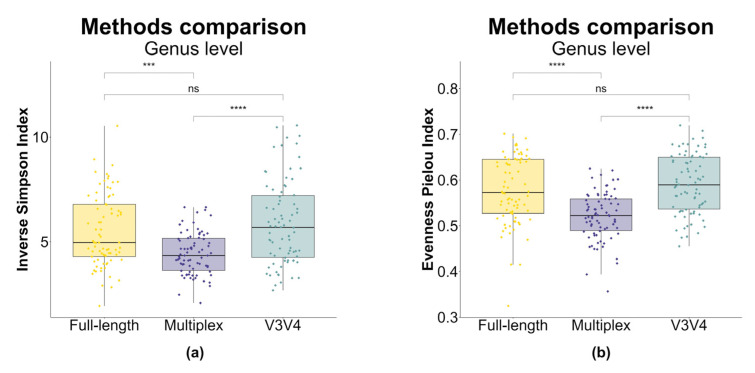
Box plot of α diversity indices calculated for each sequencing method at genus-level counts. (**a**) α diversity measured using the inverse Simpson index. (**b**) α diversity measured using Pielou’s evenness index. α diversity scores were calculated by using rarefied genus-level counts for each approach. Box plots and points represent the overall data distribution and samples, respectively. Yellow: full-length approach; violet: multiplex approach; water-green: V3V4 approach. The group means comparison was performed by using the paired Student’s *t*-test (“ns”: *p*-value > 0.05; “***”: *p*-value < 0.001; “****”: *p*-value < 0.0001).

**Figure 6 genes-14-01567-f006:**
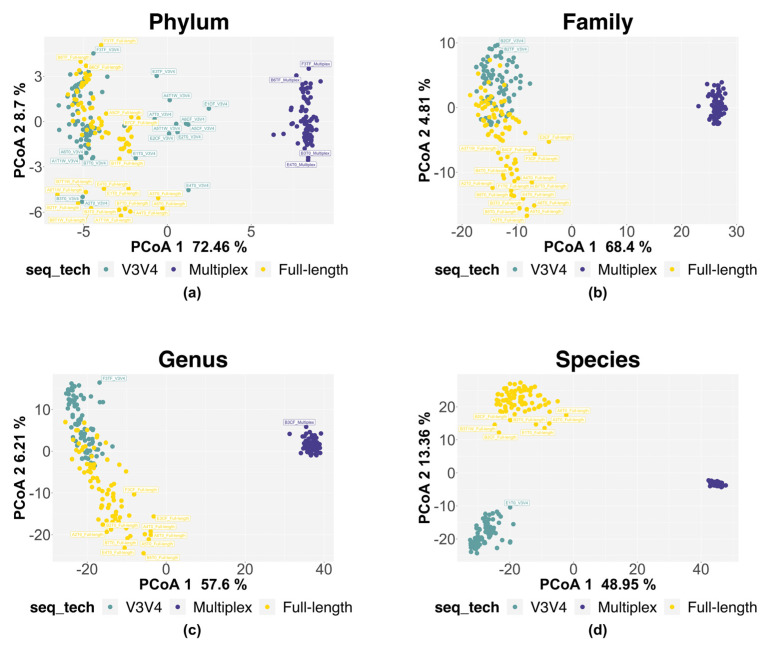
PCoA plot for taxonomic-level data. The figure shows the β diversity at different taxonomic levels, from phylum (**a**) to family (**b**), genus (**c**), and species (**d**) levels (Aitchison distance, using CLR-transformed sample abundances). Point colors represent the three sequencing method sets: the full-length approach in yellow; the multiplex approach in violet; the V3V4 approach in water-green, respectively.

**Figure 7 genes-14-01567-f007:**
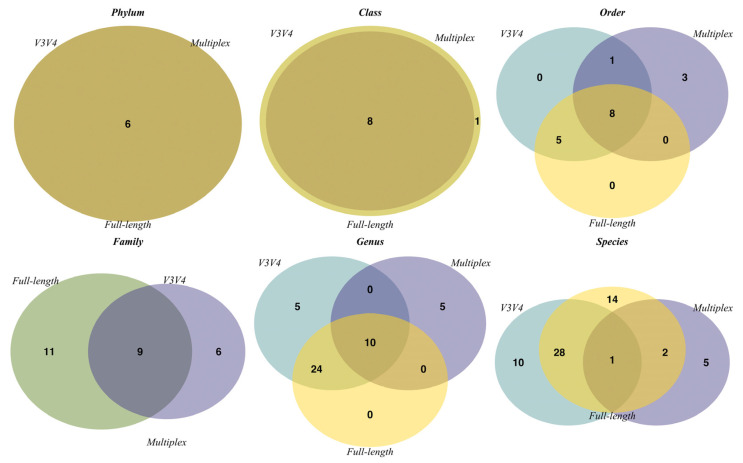
Venn diagrams using filtered taxa with relative abundance ≥ 1%. Circular colors represent the three sequencing method sets: the full-length approach in yellow; the multiplex approach in violet; the V3V4 approach in water-green, respectively. Shared taxa are represented as overlapping circles with merged colors.

**Figure 8 genes-14-01567-f008:**
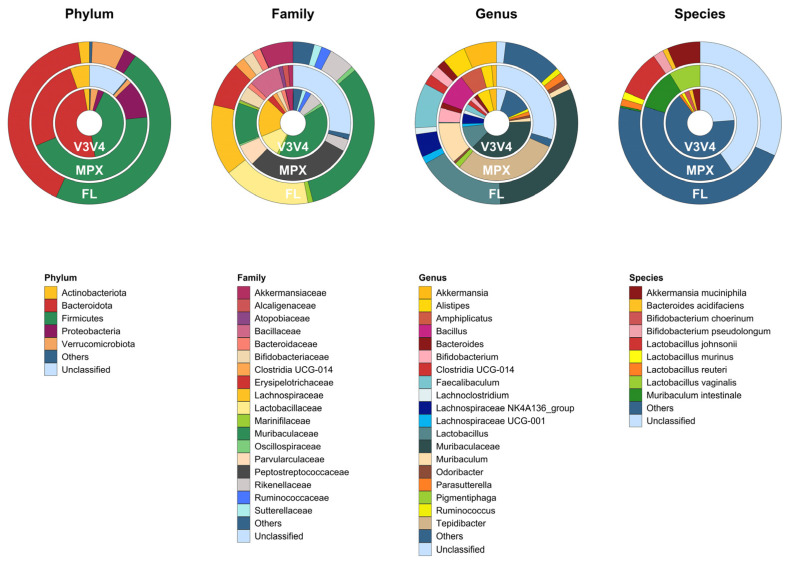
Donut charts of the taxonomic assignments at different taxonomic levels. For each sequencing method, single-region (V3V4), multiplex (MPX), and full-length (FL), the taxonomic assignments and the average relative abundances at phylum, family, genus, and species levels are plotted. Taxa with relative abundances < 1% are collapsed into “Others”.

**Figure 9 genes-14-01567-f009:**
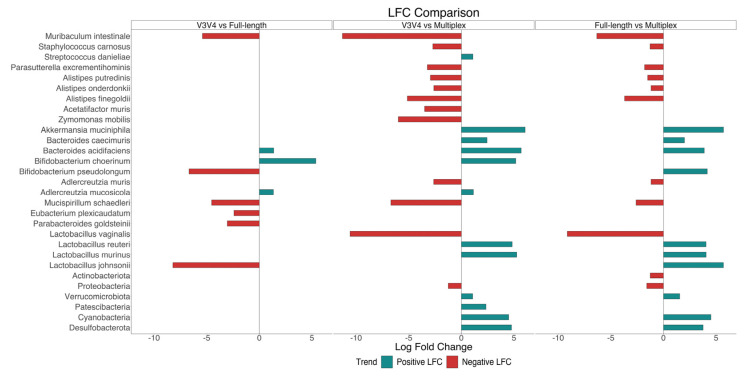
The differential abundance analysis at phylum and species levels of each sequencing method. Statistically significant differences are considered for log fold change (lfc) > |1| and adjusted *p*-value < 0.001. The pairwise comparison is performed between V3V4 vs full-length; V3V4 vs multiplex; full-length vs multiplex. The lfc value may be positive or negative if the taxa increase or decrease in the first group, respectively, compared to the second one.

**Table 1 genes-14-01567-t001:** Expected and observed genera identified by the amplicon-based sequencing approaches. Per each sequencing method, i.e., single-region (V3V4, V5V6, and V4), multiplex, and full-length, the identified genus is indicated by the + sign, otherwise the − sign is provided.

Expected Genera	Observed Genera				
	V3V4	V5V6	V4	Multiplex	Full-Length
*Enterococcus*	+	+	+	+	+
*Rhodobacter*	+	−	+	−	+
*Clostridium* ^§^	+	+	+	+	+
*Acinetobacter*	+	+	+	−	+
*Staphylococcus*	+	+	+	+	+
*Bifidobacterium*	+	+	+	+	+
*Pseudomonas*	+	+	+	+	+
*Escherichia-Shigella*	+	+	+	+	+
*Lactobacillus*	+	−	+	+	+
*Deinococcus*	+	−	+	−	+
*Neisseria*	+	−	+	−	+
*Cutibacterium*	+	−	+	−	+
*Bacillus*	+	+	+	−	+
*Helicobacter*	+	−	+	+	+
*Streptococcus*	+	+	+	+	+
*Schaalia* ^§§^	+	+	+	−	+
*Porphyromonas*	+	−	+	−	+
*Bacteroides*	+	−	+	+	+
Total observed/expected	18/18	10/18	18/18	10/18	18/18

^§^ This genus was annotated in the reference taxonomy as Clostridium senso stricto 1. ^§§^ This genus was annotated in the reference taxonomy as Actinomyces.

**Table 2 genes-14-01567-t002:** Expected and observed species identified by the amplicon-based sequencing approaches. Per each sequencing method, i.e., single-region (V3V4, V5V6, and V4), multiplex, and full-length, the identified species is indicated by the + sign, otherwise the − sign is provided.

Expected Species	Observed Species				
	V3V4	V5V6	V4	Multiplex	Full-Length
*Acinetobacter baumannii*	+	−	−	−	+
*Bacillus pacificus*	−	−	−	−	−
*Bacteroides vulgatus*	+	−	+	+	+
*Bifidobacterium adolescentis*	+	−	−	+	+
*Clostridium beijerinckii*	−	−	+	−	+
*Cutibacterium acnes*	−	−	−	−	−
*Deinococcus radiodurans*	+	−	+	−	+
*Enterococcus faecalis*	−	+	−	+	+
*Escherichia coli*	−	+	−	−	−
*Helicobacter pylori*	+	−	+	+	+
*Lactobacillus gasseri*	+	−	−	−	+
*Neisseria meningitidis*	+	−	+	−	+
*Porphyromonas gingivalis*	+	−	+	−	+
*Pseudomonas aeruginosa*	−	+	−	−	+
*Rhodobacter sphaeroides*	+	−	−	−	+
*Schaalia odontolytica*	+	−	+	−	+
*Staphylococcus aureus*	−	−	−	−	−
*Staphylococcus epidermidis*	−	+	−	−	−
*Streptococcus agalactiae*	+	−	+	−	+
*Streptococcus mutans*	+	−	+	−	+
Total observed	12/20	4/20	9/20	4/20	15/20

**Table 3 genes-14-01567-t003:** Precision, accuracy, and recall values at the genus and species levels. For each method, i.e., single-region (V3V4, V5V6, and V4), multiplex, and full-length, the values (%) correspond to the measurement at the genus and species level, considering the number of reads associated with true and false positive ASVs, true and false negative ASVs. The highest and second highest values are underlined.

	Genus			Species		
	Precision (%)	Accuracy (%)	Recall (%)	Precision (%)	Accuracy (%)	Recall (%)
V3V4	77.11	77.14	100	99.71	54.93	54.77
V5V6	33.63	29.84	71.16	1.49	1.49	4.20
V4	78.39	78.42	100	99.44	33.20	33.10
Multiplex	11.16	74.13	1.20	2.96	73.91	0.32
Full-length	83.50	83.50	100	99.95	78.10	78.11

**Table 4 genes-14-01567-t004:** Sequencing output overview for each approach and platform. For each method, (i.e., single-region V3V4, multiplex, and full-length), the final output, mean/sample reads, and mean reads length (bp) are shown.

	V3V4	Multiplex	Full-Length
Sequencing Platform	Illumina MiSeq	Illumina MiSeq	PacBio Sequel II System
Final Output (n° reads)	28,200,000	13,520,000	2,529,947
Mean read number/sample	180,769 ± 39,538	146,153 ± 57,231	28,263 ± 12,487
Mean reads length (bp)	275	250	1500

**Table 5 genes-14-01567-t005:** Permanova analysis of the taxonomic-level data. R^2^ and *p*-values represent the portion of data variability explained by the proposed model for each pairwise comparison and the achieved significance, respectively. The “Residuals” correspond to the amount of variability the model was unable to capture. Sequencing methods considered are single-region (V3V4), multiplex (MPX), and full-length (FL).

	Phylum-Level Data		Family-Level Data		Genus-Level Data		Species-Level Data	
Pairwise Comparison	R^2^ (%)	*p*-Value	R^2^ (%)	*p*-Value	R^2^ (%)	*p*-Value	R^2^ (%)	*p*-Value
V3V4 vs. FL	12.20	<0.001	11.80	<0.001	12.54	<0.001	27.65	<0.001
Residuals	87.80		88.20		87.46		72.35	
V3V4 vs. MPX	78.18	<0.001	78.23	<0.001	71.49	<0.001	71.35	<0.001
Residuals	21.82		21.77		28.51		28.65	
MPX vs. FL	74.80	<0.001	71.47	<0.001	60.63	<0.001	59.64	<0.001
Residuals	25.20		28.53		39.37		40.36	

## Data Availability

The data related to the mock benchmark analysis and the multiplex and the full-length sequencing raw data of the 78 real samples presented in this study are openly available in the SRA repository, reference number BioProject PRJNA956423. The V3V4 single-region sequencing raw data related to the 78 real samples tested for the validation are available by requesting them from the company Postbiotica s.r.l.

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
