# Peer review of "Amplicon-Based Microbiome Profiling: From Second- to Third-Generation Sequencing for Higher Taxonomic Resolution"

_genes, 2023, doi:10.3390/genes14081567_

Round 1

Reviewer 1 Report

This article compares three different amplicon sequencing approaches, first by looking at three different regions using Illumina sequencing, then multiplexing regions, and finally, full-length with PacBio sequencing.

The experimental design is sound, but the manuscript needs substantial revision before publication.

One topic of concern is that the authors need to clarify the technical limitations of the different sequencing approaches with the limitations of the taxonomic database. Because they use a mock community with a known composition (reported by ATCC), their taxonomic database should include those taxa before the analysis. Therefore, saying, for example, that none of the methods can detect Schaalia and Clostridium is not true; it is just that the names of those organisms on the ATCC standard are different from the ones present in the taxonomic database used for assignments. This needs to be clarified on the interpretation because this work is not looking into biases on bioinformatic approaches or databases but on sequencing methods. This is a fundamental issue and something that the author needs to address before acceptance of this work.

Other issues are the constant mention of metagenomics in the text, while this work is amplicon sequencing, not metagenomics. Maybe the authors should use metataxonomic instead?

Introduction

The introduction should be revised to make it more straightforward. There is a need to change and edit the language of the text. In addition, here are some comments:

-       Amplicon sequencing (or metataxonomic) differs from metagenomics, so the definition on line 52 needs to be corrected.

-       No need to capitalize amplicon sequencing in all the text.

-        On line 69, there is a mention of a lower risk of false positives. Is there a reference to support this statement?

-       The statement on lines 90-93 regarding the limitation of long-read studies needs to be revised. Several recent publications are using PacBio and ONT sequencing approaches.

Materials and Methods

-       More details are needed. For example, the authors mention a company and reference a paper for DNA extraction. This means that the DNA was extracted according to those methods?

-       Need to include information on the kit used for Illumina sequencing.

-       Bioinformatics methods need more details. The authors should include the parameters for filtering, denoising, and taxonomic assignments.

-       The same should be done for the murine biological samples. The ASVs were decontaminated using the negative control? (this needs to be clarified from the text). 

-       Any chimera checks were performed on the dataset? 

Results

One of the main comments from the results is that the authors need to change the font used for the plots. In the current version, reading and interpreting the text on the plots is challenging due to the font used.

The overall results are sound, but the text needs to be revised and checked for grammar and typos. Some additional comments:

-       Line 340, Validation on metagenomic samples. This is incorrect, as this study did not perform metagenomic sequencing. This is amplicon sequencing on real samples instead of mock communities.

-       Figure 7 is empty.

Discussion

For the multiplex approach, is it possible that this approach has a higher rate of formation of chimeras?

The authors need to revise the manuscript to check for typos and increase the clarity of the text. 

Author Response

Dear reviewer,

We thank you for your useful comments and suggestions.

In attachment, you will find our point by point responses.

Reviewer 2 Report

The paper describes a comparison between three 16S amplicon sequencing and analysis methods: sequencing of one or two hypervariable regions, sequencing of multiple hypervariable regions, and sequencing the entire 16S gene using PacBio. The analysis is based on the sequencing of both mock communities and real microbiome samples. The results suggest that the sequencing of the V3-V4 region achieves better results than the V4 and V5-V6 regions, which were also tested, and that sequencing the entire 16S gene achieves better resolution than the V3-V4. The multiplex approach was inferior to the single region approaches.

Overall, the study is informative, and researchers in the field of microbiome can benefit from the results. Below are my comments on the manuscript.

1. There are two main possible sources for error when using different amplicon methods: (i) errors related to the preparation of the data (DNA extraction, primers used, sequencing, etc.), and (ii) errors related to the bioinformatics pipeline used. To my understanding, the first part of the results (Fig. 1,2, Table 1,2) reports results that depend on both the amplicon sequencing approach and the bioinformatics pipeline, so it is impossible to know what is the source of the errors. Later the authors report on analyses that are based on the mapping of ASVs to the genomes in the mock community, which is the right thing to do in order to eliminate errors that result from the bioinformatics pipeline and focus on errors related to the amplicon technology. Still, it is not possible to know how well the results fit the proportions in the mock community in each method.

I suggest that the authors create figures that are similar to Fig. 1 and 2 and tables that are similar to Table 1 and 2 based on the mapping of the raw reads to the 20 reference genomes. In other words: calculate the fraction of reads that map to each genome for each method (observed) and compare it to the true abundances. I am not sure how this can be done with respect to the multiplex method; maybe the best thing would be to rely on the provided pipeline (see next point). It should be made clear that the results presented in the current Fig. 1, 2 and Table 1, 2 contain issues related to both the amplicon technology and bioinformatics pipeline.

2. The analysis of the multiplex method is not clear. To begin with, what is the product of the bioinformatics pipeline – an occurrence table, representative sequences, or something else? Assuming that it is an occurrence table, how was this “translated” to SILVA’s taxonomy? What is the meaning of ASVs in the context of the multiplex method – is it a single variable region (and if so, how was the rarefaction done?), a composition of different variable regions that represent a 16S gene, or something else? The methods and the results should be clear as to what exactly is the data that is evaluated.

3. When calculating alpha and beta diversities, all of the samples should be rarefied to the same depth. As done in the current study, more ASVs will be detected when rarefying to 100K reads compared to 7K. Please correct that by rarefying all the samples to the same depth, or explain why it is correct to leave it the way it is described in the manuscript.

4. In line 600 (Discussion), the rate of unclassified sequences is discussed. This should also appear in the results and not only in the figures.

Minor comments:

5. In line 125, it is unclear how much DNA was used.

2. Line 170: which implementation of dada2 was used? Was it in R or in QIIME2? What parameters were used?

6. Line 447: what is a statistically significant abundance?

7. Figure 7: the venn diagrams are not visible in the two

different pdf readers I tried. 

8. Figure 8: the names of the taxa are small and blurry. Consider

making the font bigger and the resolution higher. Also, draw the circles at the

same height. 

9. Figure 9: The title with the comparison names and the legend

titles are small and blurry. 

10. The sentence starting in line 550 is unclear but seems

important. 

11. Fig. 1-2: please add information about R^2 and not just R, to evaluate the dispersion of the points from the linear regression line.

12. Fig. 1-2 and Table S2: what is the assignment of reads that are not assigned to the expected 16S sequence?

13. Table S1: please replace the genus column (which is easy to deduce from the species name) with the phylum. This information is relevant for evaluating the primer range (and appears in the main text).

14. Table S3 is not helpful; please replace it with a summary of the number of ASVs for each taxon

15. Title for Section 3.2: please replace with “Validation on real microbiome samples”. Metagenomics usually refers to shotgun sequencing, and the use of the term in the context of amplicon sequencing is confusing.

16. Subsection 3.2.1: the number of ASVs for multiplex (52,534) is not comparable to the other two methods, probably because each 16S sequence is represented by a number of ASVs. Is it possible to state how many “strains” were identified in the multiplex method?

17. Please correct the font in the figures, it is hard to read

18. The study only addresses a specific type of environment (mammalian intestinal). It is unknown whether the results will be applicable also for much more complex environments such as soil. Please state that in the Discussion.

The manuscript could benefit from professional proofreading. A few examples:

- “Metagenomics, with its two major approaches… through high throughput sequencing approaches” (lines 52-54): this sentence is not clear. Genomes cannot be studied through amplicon sequencing. Please clarify and make it more accurate.

- Line 57: NGS should be next to Next Generation Sequencing

- “transfers” (lines 60-61) should be transfer

- “few” (line 73 and other locations in the text) should be “a few”

- “sensibility” (lines 82-83) should be sensitivity?

- “side” (line 84) should be hand

- “paired-ends reads” (line 170) should be paired-end reads

- “level” (line 240) should be levels

- “condition” (line 277) should be conditions

- “Moving at” (line 302) should be Moving to

Author Response

(The authors gave the same response as above.)
